# Laser Metal Deposition of Inconel 718 Alloy and As-built Mechanical Properties Compared to Casting

**DOI:** 10.3390/ma14020437

**Published:** 2021-01-17

**Authors:** Federico Mazzucato, Daniele Forni, Anna Valente, Ezio Cadoni

**Affiliations:** 1Automation Robotics and Machines Laboratory, University of Applied Sciences of Southern Switzerland, 6962 Viganello, Switzerland; anna.valente@supsi.ch; 2DynaMat SUPSI Laboratory, University of Applied Sciences of Southern Switzerland, 6850 Mendrisio, Switzerland; daniele.forni@supsi.ch (D.F.); ezio.cadoni@supsi.ch (E.C.)

**Keywords:** Laser Metal Deposition, Inconel 718, as-built, mechanical behavior, repairing, high strain rate

## Abstract

In the last years, powder-based Laser Metal Deposition (LMD) has been attracting attention as a disruptive Additive Manufacturing (AM) technique for both the fabrication and restoration of Inconel 718 components, enabling to overcome current limitations faced by conventional manufacturing processes in terms of manufacturing costs, tool wear, and lead time. Nevertheless, the uncertainty related to the final mechanical performance of the as-built LMD parts limits a wider adoption of such technology at industrial level. This research work focuses on the mechanical characterization of as-built Inconel 718 specimens through split Hopkinson tensile bar tests performed at different strain rate conditions. The influence of laser power on the final mechanical behavior of the as-built tensile samples is discussed and compared with the mechanical response of as-cast ones. The as-built specimens exhibit a high internal density (i.e., 99.92% and 99.90% for 300 W and 400 W, respectively) and a more ductile behavior compared to the as-cast ones for every evaluated strain rate condition. The strain hardening capacity of the as-built samples increases with the laser power involved in the LMD process, reaching an average Yield Strength of 703 MPa for specimens realized at 400 W and tested at 800/s.

## 1. Introduction

Inconel 718 is a Ni-based Superalloy specifically designed to endure severe operating conditions that demand significant mechanical resistance to critical dynamic loads applied under harsh working environments [1]. Excellent toughness, good ductility, and high creep-rupture resistance at elevated temperatures are just some advantages that encourage Inconel 718 as a reference choice for the fabrication of industrial gas turbines, jet engine components, critical rotating parts, supporting structures and pressure vessels in industrial sectors such as Aerospace and Oil and Gas [2,3]. Nevertheless, due to its superior mechanical properties, Inconel 718 is a hard-to-cut material to be machined through conventional subtractive technologies (e.g., milling, turning), requiring significant industrial resources and high production costs in terms of both tool wear, material waste (i.e., up to 95% [4]) and lead-time. In recent years, the Direct Energy Deposition (DED) technology knows as powder-based Laser Metal Deposition (LMD) has been recognized to be a cost-effective Additive Manufacturing (AM) technique for both the fabrication and refurbishment of hard-to-cut metal components [5]. Contrary to the conventional subtractive technologies, the manufacture of LMD parts takes place by adding material layer by layer through the injection of fresh metal powder particles into a locally created melt pool by means of a constant interaction between a high intensity laser source and a solid metal surface. When the melt pool moves, a solid metal bead takes form thanks to the rapid cooling of the molten material, allowing a sound bonding between the as deposited material and the underlying surface [6]. The absence of cutting tools and the limited Heat Affected Zone (HAZ) generated during the deposition process allow LMD to be particularly suitable for the manufacturing and repairing of Inconel 718 components, enabling the production of optimized and complex industrially driven geometrical shapes by means of a more efficient management and use of the raw material [5]. Despite such great technological advantages, both the high thermal gradients and the rapid cooling rates involved in the LMD process have the potential to deteriorate the final quality of the as-built part, inducing internal defects (e.g., internal cracks and keyholes), undesired geometrical distortions, or detrimental residual stresses affecting both the structural integrity and the mechanical performance of the manufactured part [7]. Several Authors have demonstrated the feasibility of LMD in processing Inconel 718 for the production of defect-free metal parts [8,9,10]. Zhong et al. [11] have analyzed the influence of laser power and metal powder on the structural integrity of Inconel 718 single linear depositions. Their work has highlighted a strong dependence of the final part quality on both the involved laser power and raw powder properties, observing a strong reduction in the formation of internal defects such as porosity, lack of fusion, and detrimental cracks when spherical-shaped powder particles and high levels of laser energy are involved. Kong et al. [12] have focused on the optimization of a LMD procedure for complex Inconel 718 depositions. Their research work has identified an optimized range of heat inputs suitable to realize defect-free Inconel 718 boss components exhibiting an acceptable internal porosity and no structural cracks, demonstrating the feasibility of LMD in realizing bulk Inconel 718 components on pre-existing curved surfaces.

Because of the thermal gradients and the rapid cooling rates involved in part manufacturing, as-built Inconel 718 LMD parts generally exhibit an anisotropic microstructure and undesired segregations that have a relevant influence on the final mechanical response of the realized components. Heat treatments performed after LMD part manufacturing have demonstrated to be an effective means to improve both mechanical performance and homogenize the final microstructure of the as-built part [13]. Blackwell [14] has analyzed the effect of the Hot Isostatic Pressure (HIP) on the final properties of 170 mm × 40 mm × 30 mm Inconel 718 blocks realized through LENS^TM^ technology. A consistent reduction in the microstructure anisotropy and an improved metal bonding between successive layers as well as an enhanced homogenization of the final mechanical properties have been obtained after HIP, removing any evidence of poor interlayer bonding. Further, Sui et al. [15] have improved the final mechanical properties of as-built LMD samples, dissolving the sharp corners and grooves of the Laves phase through solution heat treatment. Zhang et al. [16] have analyzed the quasi-static mechanical properties at different working temperatures of LMD Inconel 718 tensile samples after performing a full heat treatment (i.e., 980STA treatment and double aging). The experimental results have highlighted an enhancement of ambient temperature tensile strength from 953 to 1334 MPa after the designed thermal treatment, whereas the mechanical response of the samples at elevated temperature has been detected to vary with the building direction.

Despite heat treatments have proved to benefit from the final properties of as-built Inconel 718 components, their implementation is not always feasible. For LMD applications involving part refurbishment or surface repairing, the high temperatures, usually required for the homogenization of the microstructure and the dissolution of segregations, can deteriorate the wrought material and degrade the overall mechanical performance of the repaired part [17]. For this reason, as-built defect-free parts with acceptable mechanical properties are desirable without the implementation of post heat treatments. There are several works dealing with the mechanical behavior of as-built Inconel 718 components at high working temperatures or at high-cycling loads, but they are mainly focused on metal parts realized through Selective Laser Melting (SLM) [18,19]. Konečná et al. [20] have studied the high-cycle fatigue behavior of SLM specimens along the three spatial directions. The resulting fatigue S/N data have demonstrated that the dynamic response of the as-built specimens strongly depends on the building orientation assumed during part manufacturing, detecting the lowest fatigue strength for the specimens realized along the build direction. Further, Smith et al. [21] have focused on the influence of material anisotropy on the final mechanical properties of as-built high-density (i.e., 99.82%) Inconel 718 samples, observing a 7% decrease of the Yield strength for the vertically built specimens. To the best of author’s knowledge, only few recent works deal with the mechanical characterization of as-built Inconel 718 LMD parts. Sui et al. [17] have analyzed the mechanical response of LMD as-built samples under dynamic loads. They have detected a material mechanical performance strongly dependent on the amplitude of the external applied stress. The reason for such behavior has been correlated with the shape and size of the Laves phase detected into the as-built sample microstructure, which acts as a reinforcement hindering crack propagation for small stress amplitudes and as a crack initiation for high stress amplitudes. Yuan et al. [22] have focused on the mechanical characterization of as-built Inconel 718 samples, performing uniaxial compressive tests under high-temperature (298–1273 K) and high-strain rate (0.001–5300/s) conditions, detecting higher mechanical strengths for the manufactured samples that are parallel to the build platform compared to the perpendicular ones, due to the strong microstructure anisotropy observed after sample manufacturing. To date, the knowledge of the mechanical behavior of as-built Inconel 718 parts fabricated by LMD is still limited and little or no attention has been dedicated to the tensile mechanical behavior of as-built Inconel 718 LMD parts tested at high strain rate conditions.

For this reason, in this work, the authors focus on the influence of LMD process parameters on the final mechanical behavior of as-built Inconel 718 components tested at room temperature and at different strain rates (i.e., 0.001, 200, and 800/s) by means of a^-^ split Hopkinson tensile bar. The experimental results demonstrate a strong influence of LMD process parameters on the as-deposited material performance, observing a good experimental repeatability and a relevant mechanical response of the as-built material compared to the as-cast one. Section 2 explains the metal material and the equipment used to perform both the experimental campaign and the sample characterization, discussing the experimental methodology followed for LMD process parameters characterization and definition. Section 3 introduces and discusses the obtained results. Finally, Section 4 presents the main experimental outcomes.

The present research work presents the first scientific achievements related to the SUPSI-funded project named DED-In718, aiming to analyze the feasibility and develop the performance of LMD technology for the refurbishment and reuse of damaged Inconel 718 structural components for Aerospace applications.

## 2. Materials and Methods

### 2.1. Metal Material and LMD Equipment

The metal powder involved in the experimental campaign was an Inconel 718 gas atomized powder (supplier: LPW Technology Ltd., Philadelphia, PA, USA—see Table 1) with a grain size ranging between 44 and 106 µm (sieve analysis: Dv(10) = 43 µm; Dv(50) = 60 µm; Dv(90) = 84 µm). Circular Inconel 718 plates with a diameter of 100 mm and a thickness of 10 mm were used as both metal build platforms for LMD depositions and raw material for as-cast tensile sample manufacturing (see Figure 1).

The LMD system was a three-axis Laserdyne 430 laser cutting machine (supplier: Prima Power, Turin, Italy) retrofitted and adopted in order to perform powder-based material depositions (see Figure 2a). The system was equipped with a 1000 W fibre laser source (i.e., Convergent Photonics CF1000 (supplier: Prima Electro, Turin, Italy) with a wavelength of 1070 nm,) and a commercial Optomec^®^ multi-nozzle deposition head (supplier: Optomec, Saint Paul, MN, USA, see Figure 2b). Powder supply was ensured by a LENS^TM^ Print Engine double-hopper powder feeding system (supplier: Optomec, Saint Paul, MN, USA) connected to the deposition head.

### 2.2. Experimental Campaign

The experimental campaign was designed in order to both identify suitable combinations of process parameters for proper Inconel 718 depositions, and characterize the mechanical behavior of as-built specimens through the split Hopkinson tensile bar test. The entire experimentation was structured in three main steps (see Figure 3): characterization of the influence of the main process parameters (i.e., laser power, axis speed, and track overlap) on the final quality of simple depositions and identification of suitable process recipes for tensile sample manufacturing (step 1 of Figure 3);investigation of the influence of the chosen process recipes on the structural integrity of the realized specimens (step 2 of Figure 3); mechanical characterization of the as-built Inconel 718 samples (step 3 of Figure 3). 

The effect of laser power (W), axis speed (mm/min), and single track overlapping (%) was investigated following the LMD process design procedure discussed in [23]. Firstly, the influence of laser power (P) and axis speed (F) on the deposition accuracy and final integrity of single linear track (ST) depositions was analyzed through a full factorial Design of Experiments (DOE), taking into account 6 levels for P, 6 levels for F, and a constant powder feed rate of 0.032 g/s (see Table 2). Then, two values of ST overlap were investigated in order to characterize the as-built quality of 15 mm side square single layers (SLs). Based on preliminary experimental results coming out from the execution of preparatory depositions in conjunction with a consolidated Authors’ experience in LMD process [5,13,23,24,25], ST overlaps of 50% and 70% of the measured track width were chosen for SL manufacturing (see Table 2). For each combination of process parameters, three repetitions were performed in order to analyze process repeatability and measurement deviations.

From the ST and SL characterization, two combinations of process parameters were selected for the manufacturing of Inconel 718 vertical cylinders with a nominal height of 45 mm and a diameter of 10 mm. The vertical part orientation was chosen since the worst mechanical behavior in terms of elongation at break and tensile strength was expected along the building direction [26]. The deposition strategy followed for the printing of the cylindrical samples was a raster strategy involving a raster angle of 30 ° and a layer thickness equal to the measured average SL height. Once the deposition process was concluded, the final tensile sample geometry for split Hopkinson tensile bar test was obtained through successive post-processing operations constituted by a wire-EDM trimming followed by turning (see Figure 4). The external quality of ST and SL depositions was evaluated by means of Keyence VHX-6000 digital microscope (supplier: Keyence, Mechelen, Belgium) with a 150× magnitude lens. The ST width (w_avg_) and maximum average ST height (h_avg_) was analyzed by means of 61 profiles crossing the track length (see Figure 5), whereas the SL height (h_SL,avg_) was characterized by means of 122 profiles traced both along (i.e., 61) and transverse (i.e., 61) the laser path (see Figure 6). Since the Keyence acquisition software was not capable to automatically provide the width and the height of the realized STs and SLs, a SUPSI-developed MATLAB algorithm was designed and implemented in this analysis in order to elaborate the raw profiles coming out from microscope acquisitions (both for STs and SLs, see Figure 5c,d) and extract the required average width and height for each realized deposition (see Table 3 and Table 4). 

In order to perform the structural assessment of the samples, a complete set of STs (i.e., 36 STs) and three as-built cylinders for each evaluated combination of LMD process parameters were trimmed through wire-EDM as depicted in Figure 3—step 2 and Figure 7. Each cross-section was cold mounted in epoxy resin and polished though PRESI mecapol P320 equipment (supplier: Spectrographic Limited, Leeds, UK). Porosity size and distribution as well as melt pool depths were detected through digital image method. Three different levels of strain rate (i.e., 0.001/s, 200/s, and 800/s) were executed at an environmental temperature of 20 °C by means of split Hopkinson tensile bar in order to define the dynamic mechanical response of the as-built Inconel 718 round samples at quasi-static and high-strain rate conditions. The obtained performance was compared and assessed with the mechanical response of as-cast Inconel 718 tensile samples characterized under the same testing conditions. Three repetitions for each process and testing conditions were performed.

## 3. Results and Discussion

### 3.1. Inconel 718—LMD Process Window

Figure 8 shows the experimental outputs coming out from ST characterization and analysis. Acceptable combinations of process parameters ensuring both a proper metal bead formation and a continuous clad are represented in green. On the contrary, STs exhibiting clad instability and discontinuities are represented in red.

P mainly affects ST width and clad stability, reaching a maximum ST width of 2.2 mm for 700 W and 300 mm/min (see Table 3). When high values of P are combined with low axis speeds, a metal clad unsteadiness occurs and ST edges become unstable, exhibiting a wavy edges’ profile. Despite the presence of spattering is limited, uneven ST edges bring to irregular SL formations that are detrimental for both the structural integrity and the geometrical accuracy of the metal build, inducing internal defects (e.g., pores and cracks) and uncontrolled layer formation (e.g., undesired over-depositions) when neighboring STs are deposited. On the contrary, increasing F from 600 mm/min to 1050 mm/min, acceptable STs having a regular edges’ profile and a proper clad formation are obtained also for high levels of P (i.e., 600 W and 700 W). F exhibits a strong influence on both bead height and clad continuity. Increasing axis speed from 300 mm/min to 1050 mm/min, track heights decrease from 183 µm to 43 µm on average, whereas lack of fusion is observed when high speeds (i.e., 1050 mm/min) are combined with medium-low laser powers (i.e., 400–200 W). The effect of F on the ST metal clad integrity is more evident at 200 W. In this case, a discontinuous deposition occurs for F ranging between 750 and 1050 mm/min and a broken metal clad formation is detected (see Figure 8). The effect of F on ST continuity is explained by the discrete nature of the powder flow in conjunction with low values for both the linear mass density [7] and laser exposure time [27], respectively defined as:Linear Mass Density = f/F,(1)
Laser Exposure Time = d/F,(2)
where d is the diameter of the laser spot and f is the powder feed rate. Increasing axis speed, both (1) and (2) decrease, limiting the melt pool size and reducing its capability to catch a sufficient amount of powder particles in order to ensure a consistent and continuous deposition. Moreover, the intrinsic fluctuations in the powder flow prevent a constant supply of raw material to the melt pool over time, encouraging a not constant material deposition and lacks of fusion when high values of F are employed. This phenomenon, combined with a smaller melt pool size due to low values for both linear mass density and laser exposure time, leads to discontinuous metal clad formation and deteriorates the structural integrity of the metal build.

From the analysis of the ST cross-sections, no internal pores or cracks are observed (see Figure 9), proving that all the evaluated combinations of process parameters allow the formation of a pore-free bonding between the as-deposited material and the build surface although clad integrity is not ensured for high axis speed. Contrary to the Inconel 718 plate that shows a large amount of inclusions that can be explained as primary segregations as a consequence of the casting process employed for plates’ production [28], the ST cross-sections exhibit a limit presence of such inclusions due to the high cooling rate involved in the LMD process. The nature of the observed dark areas inside the metal bead can have a double interpretation. The presence of inclusions close to the edge between the metal bead and the Inconel 718 plate (green arrows in Figure 9b,c) can be primary segregations that moved from the plate to the metal bead due to the convective fluid-dynamic flows developed during melt pool formation and enabling the diffusion of segregations inside the metal clad cross-section. On the contrary, inclusions located more in the center of the metal bead (blue arrows in Figure 9b,c) can indicate the presence of small oxides formed during melt pool formation and cooling. The presence of oxides in Inconel 718 LMD samples has been demonstrated by several Authors [29,30] when the deposition process is performed without a closed and controlled deposition environment, indicating that the local supply of inert gas ensured by the deposition head is not sufficient to properly protect the melt pool from oxidation. Moreover, Al/Ti-rich oxides have been previously detected by Saboori et al. [13,31] in the microstructure of Inconel 718 as build samples manufactured by means of the same LMD equipment and same metal powder used for running the experimental campaign discussed in Section 2.2, proving the limitation of the current deposition head solution employed for sample manufacturing. Despite the previous experimental results, in order to confirm or reject these considerations, further analyses are running to detect the chemical compositions of the observed inclusions and fully characterize the metal bead formation. The image analysis of the melt pool cross-sections reveals high penetration depths ranging between 148 and 994 µm depending on the process parameters involved for the depositions (see Figure 10). In particular, P strongly affects melt pool dilution into the metal substrate. At 300 mm/min, the metal bead penetration increases from 188.5 µm up to 993.9 µm when P moves from 200 W to 700 W, observing a variation of dilution ratios between 60% and 90%. High axis speeds (i.e., 1050 mm/min) attenuate the effect of the laser power, detecting penetration depth ranging between 147.5 µm and 492.8 µm for 200 W and 600 W respectively. Despite dilution ratios higher than 50% could conduct to keyhole phenomena [7], no internal pores are detected.

The influence of ST overlapping on SL deposition has been analyzed by taking into account the four combinations of process parameters listed in Table 4, with the perspective to prevent part failures during the manufacturing of the Inconel 718 tensile samples (see Figure 4). Indeed, especially for parts with medium–high aspect ratios, the use of medium-low laser powers limits the generation of high thermal fluxes and detrimental overheating when compared to the application of medium–high laser energy, limiting the formation of high thermal gradients between the construction platform and the upper layers deposited and reducing the risk of part distortions during sample manufacturing. SL external analysis demonstrates that increasing ST overlapping from 50% to 70%, layer height becomes more consistent and uniform, attenuating the wavy profile of the top SL surface and reducing the average peak-to-valley distance of 43% and 49% for P 300 W and P 400 W respectively. A sharpen wave profile or the presence of strong inhomogeneities on the top surface of the SL compromises the successive deposition of a new metal layer, inducing an uncontrollable part growth during the deposition process and decreasing the geometrical accuracy of the realized part. For this reason, an ST overlapping of 70% is chosen for the manufacturing of the cylindrical tensile specimens through LMD. Table 4 summarizes both the resulting SL height for each evaluated process parameters and the LMD process recipes chosen for the manufacturing of tensile sample.

The realized cylindrical specimens show a good dimensional accuracy and no evident distortions. The structural characterization reveals an average internal density of 99.92% and 99.90% for as-built samples realized at 300 W and 400 W, respectively (see Figure 11). Due to the limitations of the image method employed for the detection and quantification of pores, it was no possible to effectively distinguish internal defects (such as pores) from intermetallic segregations and inclusions formed during part manufacturing. For this reason, the structural characterization is performed considering every detected dark area as an internal defect. Despite such an assumption, a high internal density is observed from the performed cross-section analysis. The detected defect size ranges between 17 and 59 µm for samples built at 300 W and between 14 and 62 µm for samples built at 400 W. Since no difference between pores and segregations is performed, it is reasonable to assume an actual internal density higher than the measured one, supporting the goodness of the evaluated LMD process parameters for the manufacturing of Inconel 718 parts.

### 3.2. Mechanical Behavior of As-built Inconel 718 Specimens

Figure 12 and Figure 13 show the engineering stress-strain curves obtained from split Hopkinson tensile bar testing [32,33,34]. Both the as-cast and the as-built materials exhibit a strain-sensitive behavior, showing a strain hardening capacity for all the evaluated strain rates (see Figure 12 and Table 5). Specifically, increasing the strain rate from 0.001/s (i.e., quasi-static) to 800/s (i.e., high strain rates), the Yield Strength (YS) and Ultimate Tensile Strength (UTS) of the as-cast samples move from 582 MPa and 895 MPa to 772 MPa and 1031 MPa respectively, whereas for the as-built samples realized at 400 W the YS and UTS move from 554 MPa and 869 MPa to 703 MPa and 958 MPa respectively. Both kinds of as-built samples show a weaker sensitivity to strain rate compared to the as-cast one. Indeed, increasing the strain rate conditions from quasi-static up to 800/s, the increment of YS and UTS is of, respectively, 20% and 7% for a laser power of 300 W and of 27% and 10% for a laser power of 400 W. On the contrary, the as-cast samples increase their YS and UTS of 33% and 15% respectively when the strain rate is increased from 0.001/s up to 800/s. The different sensitivity to strain rate between the as-built and as-cast samples can be explained by the two diverse expected microstructures resulting from the two manufacturing processes which involve different cooling rates. As stated by Urdanpilleta et al. [35], grain boundary sliding is recognized as one of the main mechanisms driving the strain rate sensitivity behavior of Inconel 718. In the as-cast samples, the resulting microstructure is expected to be more homogeneous compared to the as-built ones due to the lower cooling rate, exhibiting equiaxed grain formation with no preferred growth direction [36,37]. On the contrary, in the as-built samples due to the higher cooling rates and the unidirectional deposition, the resulting microstructure is widely accepted and demonstrated to be anisotropic and mainly composed by columnar dendrites along the building direction [38]. As reported by Yuan et al. [39], the presence of dendrite arms affects the mechanical response of the as-built material at high strain rate, reducing the flow stress and the hardening of the alloy for external loads applied along the building direction, promoting the sliding between neighboring grains and decreasing the strain rate sensitivity of the as-built Inconel 718 samples compared to the as-cast ones.

The experimental outputs highlight an improved repeatability and a more ductile behavior of the as-built LMD material compared to the as-cast specimens, exhibiting higher elongations at break and reaching an elongation of 51.6% and 45.9% (i.e., 300 W and 400 W, respectively) at 20 °C and 800/s of strain rate (see Table 5). Moreover, the mechanical behavior of the as-built material show a dependence on the combinations of process parameters involved in LMD manufacturing. Indeed, for each evaluated testing condition, the as-built samples realized with a laser power of 300 W exhibit the lower mechanical strength in terms of UTS and the higher elongation at break, detecting an average 10.4% increase of the uniform strain compared to the as-built specimens realized at 400 W (see Table 5). On the contrary, the as-built specimens realized at 400 W and 750 mm/min show a very similar behavior in terms of both UTS and uniform elongation in comparison to the as-cast material, as depicted in Figure 13 and Table 5.

The reason for the higher mechanical strength and the lower elongation at break detected for the as-built specimens realized at 400 W can be explained by an increased presence of inclusions compared to the samples built at 300 W, which appear as bright areas in Figure 14. As stated by several works [29,30,40], the microstructure of Inconel 718 parts produced by LMD can exhibit both the presence of inclusions of oxides and the formation of intermetallic segregations due to the high thermal gradients involved during the deposition process. Moreover, the employment of high energy inputs in LMD have been demonstrated to facilitate the formation and the growth of Laves phase and carbides into the as-deposited Inconel 718 microstructure due to the low cooling rates involved [13,41,42]. The Laves phase is a hard and brittle intermetallic compound and, in conjunction with carbides, affects the final performance of Inconel 718. Several authors have demonstrated the influence of both the shape and the distribution of the Laves phase and carbides on the final properties of Inconel 718 parts, demonstrating that an excessive presence of intermetallic compounds decreases the final mechanical properties of the as-built parts, lowering both the material strength and the elongation at break and inducing a more brittle behavior of as-built specimens compared to wrought material. Nevertheless, other authors have proved that a controlled presence of the Laves phases acts as a hardening phase for Inconel 718 LMD parts, increasing the UTS of the as-built specimen tested at environmental temperature and exhibiting higher mechanical properties compared to the wrought material [17]. Based on the previous aforementioned research works on Inconel 718 parts realized through LMD, it is reasonable to assume that the different mechanical performance observed between the as-built samples realized at dissimilar laser powers depends on a different size and distribution of segregations generated during the deposition process, which hinder the plastic deformation and block the movement of dislocation [39]. Indeed, the specimens realized with a global energy density (GED [7]) of 24 J/mm^2^ (i.e., 300 W) exhibit the lower UTS and a longer elongation at break in conjunction with both a lower presence and a lower size (i.e., between 17 and 59 µm) of segregations due to the lower cooling rates involved in part manufacturing. On the contrary, increasing the energy input (i.e., GED = 32 J/mm^2^) both the presence and the size (i.e., between 14 and 62 µm) of segregation increases, resulting in an improved final mechanical behavior of the realized specimens. In order to validate such assumptions, a further analysis on the resulting final microstructure and chemical element distribution is running with the perspective to better explain the final mechanical properties of the as-built samples and their comparison with the as-cast ones. Despite such considerations, the good repeatability of the experimental results coming out from split Hopkinson tensile bar testing demonstrates the robustness of the combinations of LMD process parameters involved for LMD sample manufacturing.

The Dynamic Increase Factors (DIFs) for both the YS (DIF fy) and the UTS (DIF fu) are reported in Figure 15 as a function of the strain rates. DIFs are evaluated as the ratio between the dynamic and the quasi-static values and it is possible to observe higher DIFs for the YS compared to the UTS for each tested sample and every evaluated testing condition. Moreover, both the as-built LMD materials exhibit a DIFs’ trends similar to the as-cast ones with the exception of yield data at higher strain rates, proving the weaker strain rate sensitivity of the as-built samples and confirming the mechanical behavior observed in Figure 12 and Figure 13.

The ratio between the UTS to the YS (fu/fy) as well as the uniform and failure strains are reported in Figure 16.. These parameters describe the material behavior in the plastic zone in terms of ductility. Similar trends for the as-cast and the as-built materials can be observed since the fu/fy ratio decreases with increasing strain rate, meaning a reduction in ductility. Nevertheless, for the as-cast material, the trend of fu/fy in relation to the observed effective strain rate underlines a sharper increase of the Yield stress compared to the UTS with the strain rate (see Table 5), highlighting a higher strain rate sensitivity of the as-cast samples compared to the as-built ones. For both the as-built materials, the evolution of the uniform strain underlines a slightly constant behavior for strain rates ranging between 200 and 800/s, whereas, the elongation at failure increases, confirming both the lower strain rate sensitivity and the increased ductility detected in Figure 12 and Figure 13.

## 4. Conclusions

In this research work, the influence of the LMD process parameters on the final mechanical properties of as-built Inconel 718 specimens is analyzed and discussed, providing a proper process window for defect-free Inconel 718 depositions for the repairing of damaged metal surfaces. The main research outputs are:Inconel 718 exhibits a wide processability window with respect to the range of evaluated LMD process parameters, allowing the formation of proper metal beads with deep melt pool penetration depths to be involved in the refurbishment of worn Inconel 718 components;the as-built specimens exhibit good mechanical properties with a more ductile behavior compared to as-cast ones;the as-built specimens show a weak sensitivity to strain rate for both combinations of process parameters employed in the experimental campaign, detecting a smaller increment in both DIFf_y_ and DIFf_u_ compared to the as-cast ones for strain rates ranging between 200 and 800/s;increasing laser power up to 400 W, the measured UTS increases for all the evaluated testing conditions, detecting a mechanical behavior similar to the as-cast one;the increase in UTS and decrease in elongation at break detected for the as-built specimens at 400 W can be explained by a higher presence of inclusions compared to the as-built specimens at 300 W. Nevertheless, further analyses are required to properly characterize the mechanical behavior of the as-built samples and a microstructural characterization is currently running in order to identify and quantify the metallic inclusions detected by optical imaging;experimental observations point out that the achievement of suitable mechanical properties is possible without the implementation of post-heat treatments;the achieved research outputs demonstrate the feasibility to involve LMD for both the refurbishment of Inconel 718 worn parts and the production of Inconel 718 high-aspect-ratio features with acceptable as-built final mechanical properties which are comparable to the as-cast specimens.

## Figures and Tables

**Figure 1 materials-14-00437-f001:**
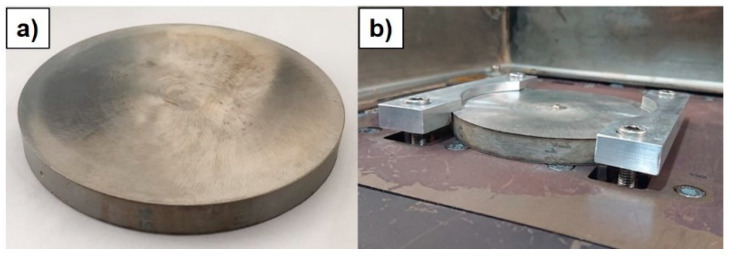
(**a**) circular Inconel 718 build platform used for LMD depositions; (**b**) Inconel 718 build platform clamped to the LMD workpiece table.

**Figure 2 materials-14-00437-f002:**
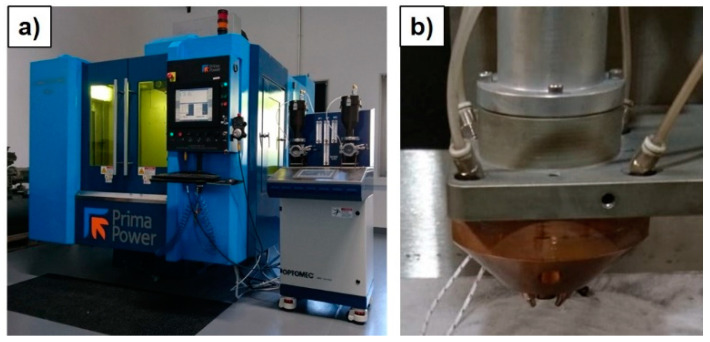
(**a**) Lasedyne 430 machine; (**b**) multi-nozzle Optomec^®^ deposition head.

**Figure 3 materials-14-00437-f003:**
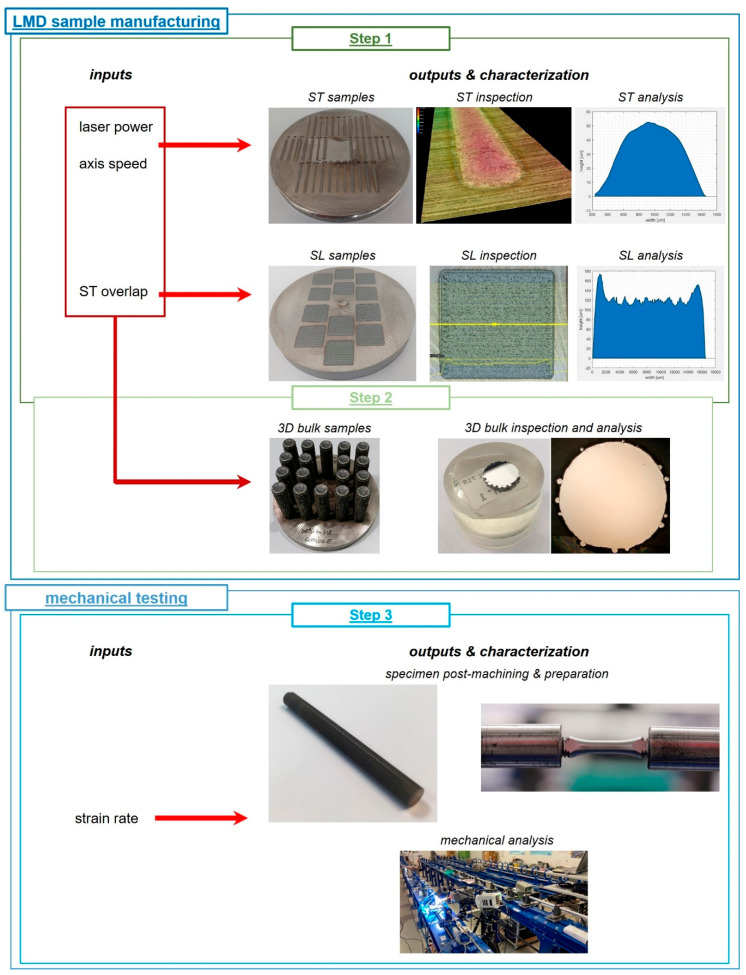
Structure of the designed experimental campaign.

**Figure 4 materials-14-00437-f004:**
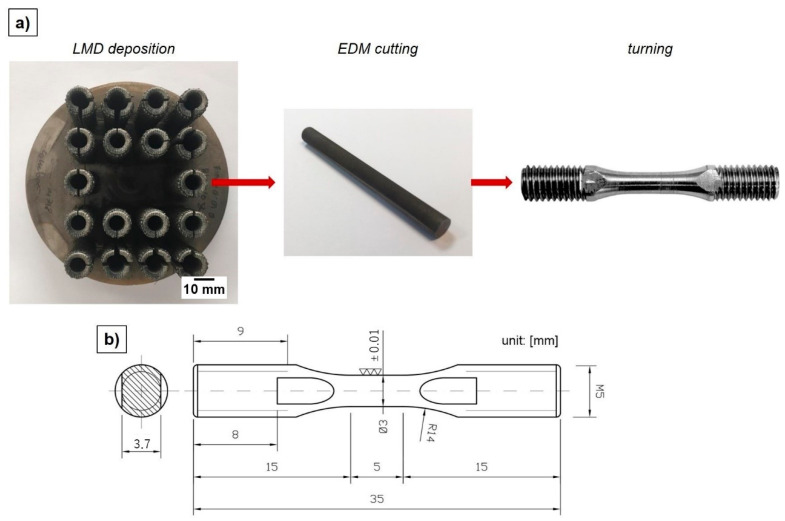
(**a**) tensile sample preparation from as-built Inconel 718 cylinders; (**b**) final tensile sample size and geometry.

**Figure 5 materials-14-00437-f005:**
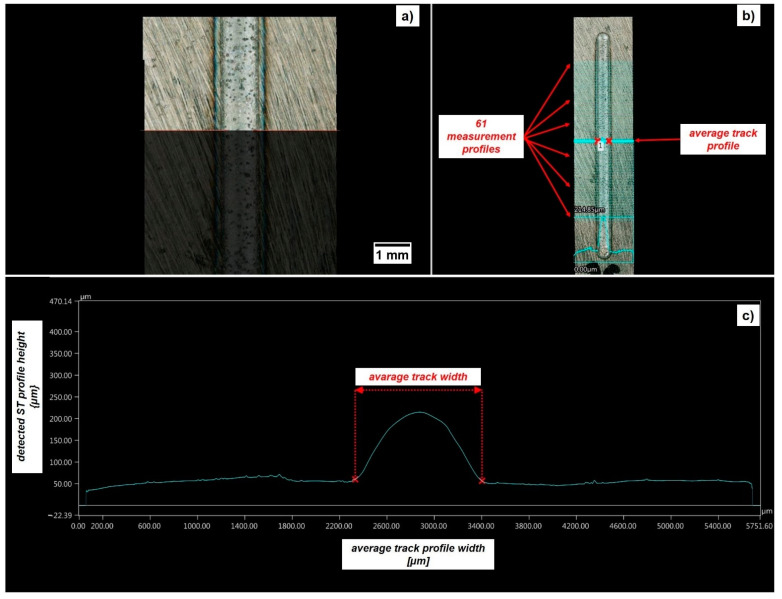
Single Track (ST) acquisition and characterization: (**a**) Inconel 718 ST acquisition; (**b**) tracing of 61 profiles crossing the ST length; (**c**) average ST profile detection.

**Figure 6 materials-14-00437-f006:**
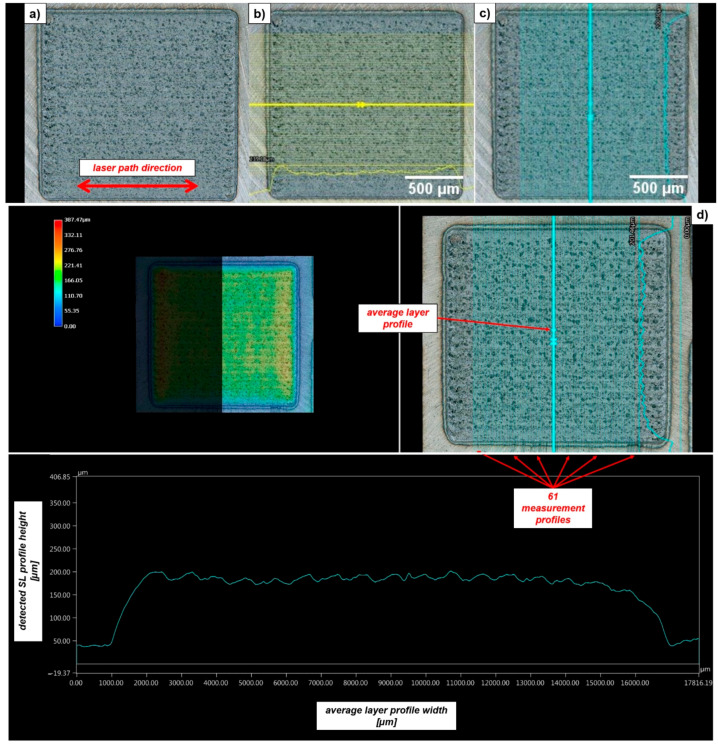
SL acquisition and characterization: (**a**) Inconel 718 SL acquisition; (**b**) tracing of 61 profiles along the laser path direction; (**c**) tracing of 61 profiles transverse to the laser path direction; (**d**) average SL profile detection for the height measurement (transverse to the laser path direction).

**Figure 7 materials-14-00437-f007:**
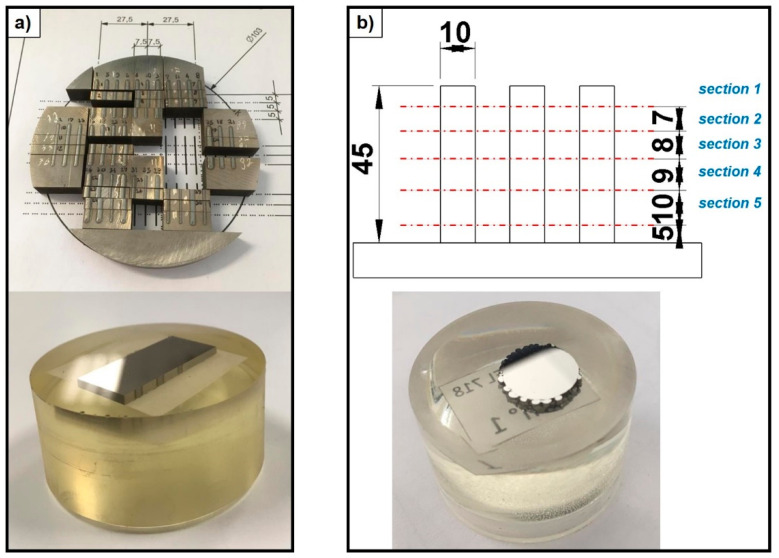
(**a**) ST cuts and cross-sections (mm); (**b**) cut positions for the structural analysis of as-built Inconel 718 cylinders and as-built cylinder cross-section (mm).

**Figure 8 materials-14-00437-f008:**
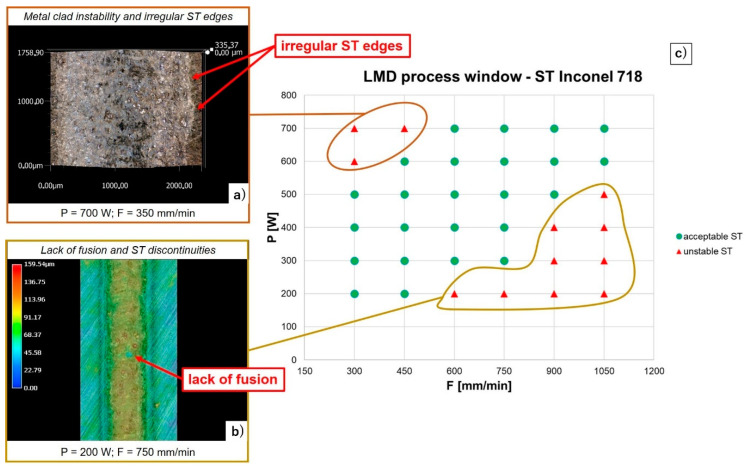
LMD process window for Inconel 718 STs: (**a**) irregular ST edges due to metal clad instability; (**b**) lack of fusion; (**c**) ST results and characterization.

**Figure 9 materials-14-00437-f009:**
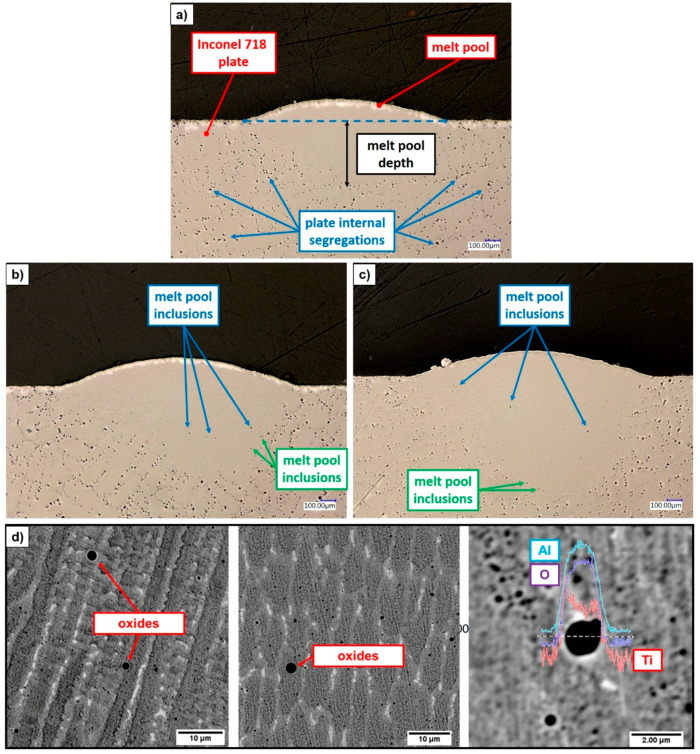
ST cross-sections: (**a**) 300 W, 300 mm/min; (**b**) 400 W, 300 mm/min; (**c**) 500 W, 300 mm/min; (**d**) detected Al/Ti-rich oxides in Inconel 718 as deposited material (SEM analysis) [31].

**Figure 10 materials-14-00437-f010:**
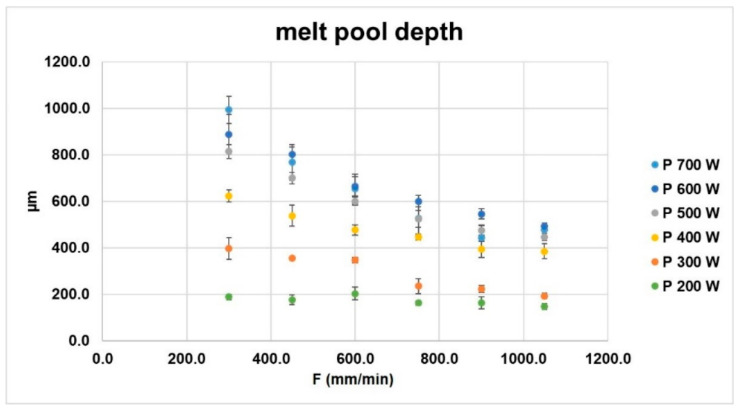
Detected melt pool depth.

**Figure 11 materials-14-00437-f011:**
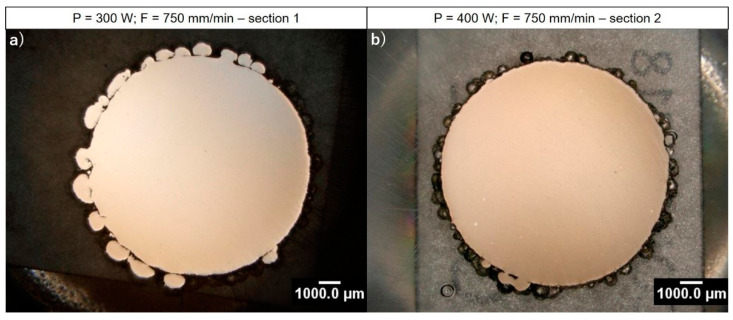
Structural analysis of the as-built cylinders cross-sections through optical microscope: (**a**) cylinder built at P 300 W and F 750 mm/min; (**b**) cylinder built at P 400 W and F 750 mm/min.

**Figure 12 materials-14-00437-f012:**
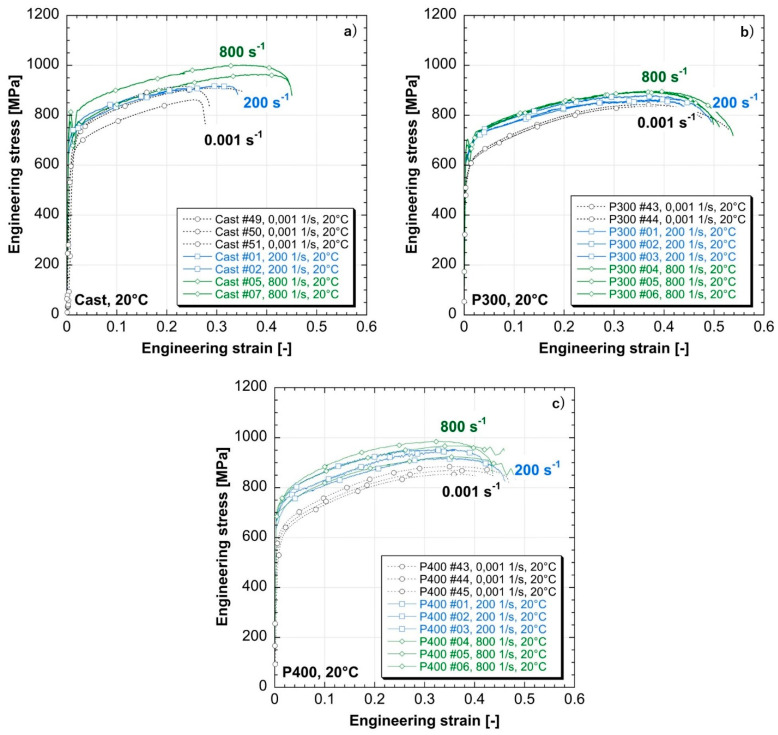
Mechanical behavior of the realized specimens at different strain rates: (**a**) as-cast; (**b**) as-built at 300 W; (**c**) as-built at 400 W.

**Figure 13 materials-14-00437-f013:**
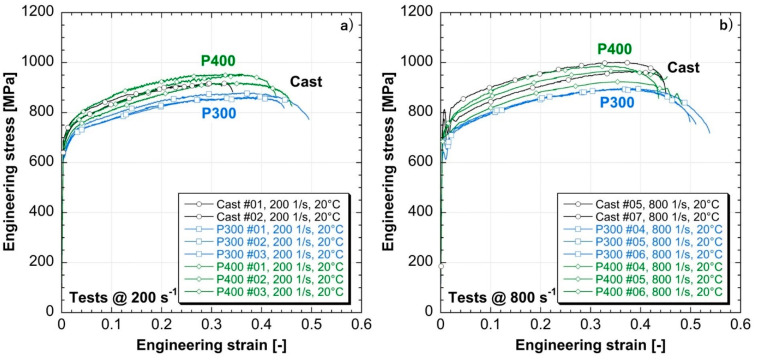
Mechanical tensile properties of the realized specimens: (**a**) at 200/s; (**b**) at 800/s.

**Figure 14 materials-14-00437-f014:**
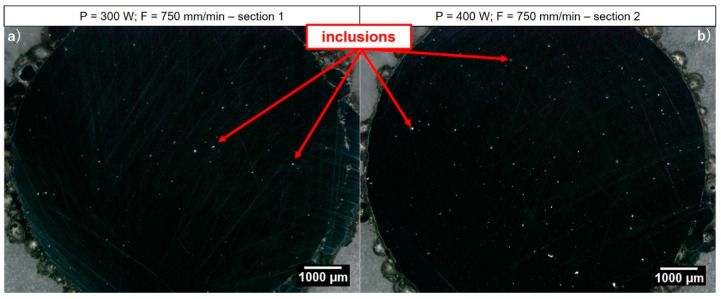
Segregation distribution on the cross-section of as-built Inconel 718 specimens: (**a**) cylinder built at P 300 W and F 750 mm/min; (**b**) cylinder built at P 400 W and F 750 mm/min.

**Figure 15 materials-14-00437-f015:**
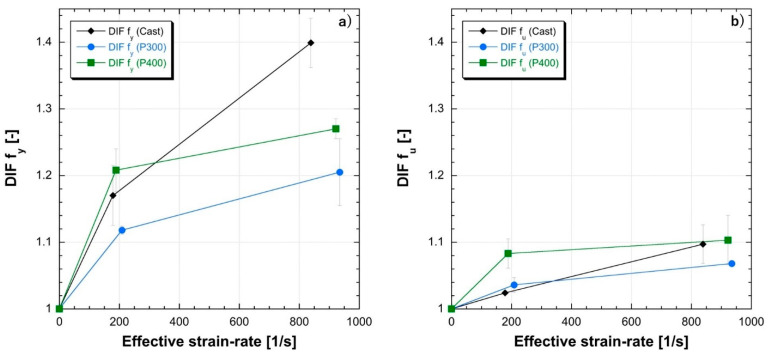
Dynamic Increase Factors for (**a**) YS (DIF f_y_) and (**b**) UTS (DIF f_u_).

**Figure 16 materials-14-00437-f016:**
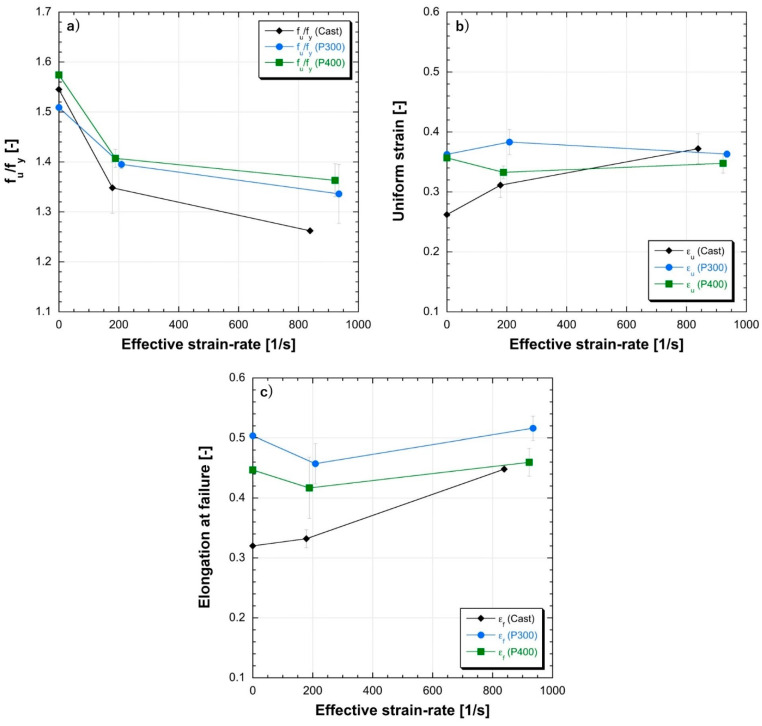
Ductility parameters of the realized specimens: (**a**) UTS / YS ratio; (**b**) uniform strain; (**c**) elongation at failure.

**Table 1 materials-14-00437-t001:** Chemical analysis of Inconel 718 powder (weight %).

C	Mn	Si	Ni	Cr	Co	Al	Mo	Nb	Fe
0.06	0.01	0.07	54.38	18.82	0.09	0.57	3.17	5.05	Bal.

**Table 2 materials-14-00437-t002:** Evaluated LMD process parameters for ST and SL depositions.

LMD Process Parameters	Min. Value	Max. Value	Number of Levels	Step Value
Laser power (W)	200	700	6	100
Axis speed (mm/min)	300	1050	6	150
ST overlap (%)	50	70	2	20

**Table 3 materials-14-00437-t003:** Detected ST width and height (average and standard deviation). “A” = acceptable ST; “L.F.” = lack of fusion; “I.E.” = irregular ST edge.

ST	P (W)	F (mm/min)	w_avg_ (µm)	w_StdDev_ (µm)	h_avg_ (µm)	h_StdDev_ (µm)
A	200	300	1339	29	125	9
A	200	450	1246	92	82	5
L.F.	200	600	1146	93	55	7
L.F.	200	750	1105	120	39	6
L.F.	200	900	1065	66	35	5
L.F.	200	1050	1030	52	29	4
A	300	300	1503	61	157	11
A	300	450	1394	61	95	10
A	300	600	1323	58	70	10
A	300	750	1239	33	49	4
L.F.	300	900	1204	44	39	3
L.F.	300	1050	1171	85	30	5
A	400	300	1723	80	173	26
A	400	450	1563	57	117	13
A	400	600	1462	52	81	4
A	400	750	1381	46	66	8
L.F.	400	900	1388	67	56	6
L.F.	400	1050	1367	35	41	9
A	500	300	1792	98	193	18
A	500	450	1805	136	116	17
A	500	600	1662	39	87	10
A	500	750	1671	171	74	9
A	500	900	1473	84	57	9
L.F	500	1050	1482	50	48	9
I.E.	600	300	1838	35	208	18
A	600	450	1961	178	148	8
A	600	600	1828	82	97	7
A	600	750	1744	85	76	9
A	600	900	1674	88	65	12
A	600	1050	1671	114	52	7
IE.	700	300	2208	71	244	10
I.E.	700	450	2168	112	160	11
A	700	600	2166	132	114	4
A	700	750	2089	75	84	17
A	700	900	1996	78	69	8
A	700	1050	1843	109	58	9

**Table 4 materials-14-00437-t004:** Detected SL height at different LMD combination of process parameters.

P (W)	F (mm/min)	ST Overlap (%)	h_SL,avg_ (µm)	h_StdDev_ (µm)	Peak-to-Valley Distance (avg) (µm)	LMD Recipe Involved in Tensile Sample Manufacturing
300	750	50	72	10	21	No
300	750	70	118	21	12	Yes
400	750	50	102	16	28	No
400	750	70	162	39	14	Yes

**Table 5 materials-14-00437-t005:** Mechanical performance of the as-cast and as-built specimens. Standard deviation is in brackets.

	0.001 (/s)			200 (/s)			800 (/s)		
	As-cast	As-built (P = 300 W)	As-built (P = 400 W)	As-cast	As-built (P = 300 W)	As-built (P = 400 W)	As-cast	As-built (P = 300 W)	As-built (P = 400 W)
Proof Stress (R0,2%) (MPa)	582 (53)	557 (33)	554 (42)	685 (55)	623 (3)	669 (18)	772 (19)	671 (28)	703 (8)
UTS (MPa)	895 (29)	838 (6)	869 (15)	909 (102)	869 (9)	941 (19)	1031 (87)	895 (2)	958 (32)
Uniform Strain (%)	26.2 (3.3)	36.3 (0.5)	35.7 (0.5)	27.4 (4.8)	38.3 (2.1)	33.3 (1.1)	27.5 (4.8)	36.3 (0.6)	34.8 (1.6)
Fracture stress ENG (MPa)	769 (85)	759 (31)	801 (31)	855 (103)	816 (43)	880 (63)	942 (73)	745 (24)	879 (60)
Fracture strain ENG (%)	32.0 (6.2)	50.4 (3.9)	44.7 (2.2)	29.4 (4.8)	45.7 (3.4)	41.7 (5.1)	41.6 (5.5)	51.6 (2.0)	45.9 (2.3)

## Data Availability

Data available on request due to restrictions.

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
