# Peer review of "Laser Metal Deposition of Inconel 718 Alloy and As-built Mechanical Properties Compared to Casting"

_materials, 2021, doi:10.3390/ma14020437_

Round 1

Reviewer 1 Report

This work investigates the effect of LMD process parameters on the quality and properties of Inconel 718 specimens. The manuscript is well-written, but the content is not very new, because many previous studies have reported the influence of process parameters on the quality of LMD fabricated parts. The mechanical behavior of as fabricated Inconel 718 specimens is seldomly reported, but the interesting observation of weak sensitivity to strain rate of the as-fabricated sample, compared with the cast sample is not very well explained. More discussion should be given.

The quality of Fig5 a and b seems to be not good. In addition, the lines inside the figure are not explained.

The definition of ST width and heights in Table 3 for the whole sample should be given.

The authors stated that “no internal pores are observed in Fig. 7”. Please explain what the small tiny dark voids are in Fig. 7.

The analysis about the segregation in Fig. 12 is too preliminary.

Author Response

Dear reviewer,

the Authors would like to thank you for your valuable comments and suggestions. We hope to have properly responded to your comments and to have significantly improved the quality of our Manuscript.

Please, see the attachment (the references are at the end of the document).

Thank you

Reviewer 2 Report

The paper is interesting because concerns the mechanical behaviour of Inconel 718 components built by Laser metal Deposition, however the work, in my opinion, is not clear and needs major revisions.

  • The paper seems to focus on the mechanical behaviour of cylindrical parts built by LMD, compared with the cast ones. In the abstract, authors say" This research work focuses on the mechanical characterization of as built Inconel 718 specimens through split Hopkinson tensile bar tests performed at different strain rate conditions. A suitable process window for proper LMD Inconel 718 depositions is provided..” In the introduction (lines 113-115) authors say “A structured procedure for the identification of proper LMD process parameters is introduced and discussed in order to provide a suitable process window for defect-free Inconel 718 part repairing”.

The question is: the objective of the work is the analysis of the mechanical behaviour of the 3D parts fabricated by LMD or the study of repaired parts produced by LMD?

  • In figure 3, step 1, the experimental procedure for step 1 is illustrated. It is not clear how 3D bulk samples have been fabricated. Authors should explain and clarify how they have built those samples, what kind of building strategy was used. Always in the same picture, at the 3D bulk inspection and analysis, it seems to me that samples were fabricated with a conventional process (i.e casting) and that they were only coated by LMD. This aspect again is not clear and should be clarified in all the paper.
  • The abstract should be modified reporting the main results coming from the research
  • In the introduction is not clear the innovative content of the paper in reference to the state of art. Thus, the innovative aspects of the research should be highlighted.
  • In section 2.2, Table 2 reports LMD process parameters used for experiments. How was chosen the degree of overlap ST? Authors have chosen a range of variability between 50% and 70%. The choice of these values comes from previous experiments or from scientific literature? Please justify the choice.
  • Images of Figure 7 should be improved. It is not possible to detect the dilution. Also the quality of images is poor.
  • Figures 10 and 11 are not clear. Letters inside the images are too small.

Author Response

Dear reviewer,

the Authors would like to thank you for your valuable comments and suggestions. We hope to have properly responded to your comments and to have significantly improved the quality of our Manuscript.

Please, see the attachment (the references are at the end of the document)

Thank you

Round 2

Reviewer 1 Report

1 Scale bar should be added in Fig. 5a and Fig. 5b. The meaning of the x axis and y axis is missing in Fig. 5c and Fig. 6d.

2 The authors should add the results of the chemical compositions analysis to support the conclusion about the melt pool inclusions. This is also important for the analysis of the mechanical properties.

Author Response

Dear reviewer,

the Authors would like to thank you for your valuable comments and suggestions. We hope to have properly respond to your comments and to have significantly improved the quality of our Manuscript.

Please, find the answers to your kind comments as follows.

Thank you

Point 1: Scale bar should be added in Fig. 5a and Fig. 5b. The meaning of the x axis and y axis is missing in Fig. 5c and Fig. 6d. 

 Response 1: Thank you for your comment and suggestions. Both Figure 5 and Figure 6 have been improved, specifically:

  • a scale bar has been added both in Figure 5a and Figure 5b;
  • the meaning of x and y axes has been added in Figure 5c and 6d.

Point 2: The authors should add the results of the chemical compositions analysis to support the conclusion about the melt pool inclusions. This is also important for the analysis of the mechanical properties.

Response 2: Thank you for your comment. The Authors apologise for this, but a deeper microstructural analysis in conjunction with a chemical analysis is running and the achieved experimental results are still not available for publication. Both the ST, SL, and 3D-bulk cross-sections have been sent for microstructural characterization, but at the moment no results are available to the Authors. In order to better support the Authors’ considerations concerning melt pool inclusions, previous results in terms of Al7Ti-rich oxides have been added in the Manuscript (lines 264-268 and Figure 9d of the new version of the Manuscript). As discussed in the text, this kind of oxides have been detected in Inconel 718 as built components realized in the framework of EU H2020 4D-Hybrid project, employing the same LMD system and same metal powder employed for the depositions realized in the Manuscript. 

Lines 264-268

“Moreover, Al/Ti-rich oxides have been previously detected by Saboori et al. [13,31] in the microstructure of Inconel 718 as build samples manufactured by means of the same LMD equipment and same metal powder used for running the experimental campaign discussed in Section 2.2, proving the limitation of the current deposition head solution employed for sample manufacturing.”

Moreover, the conclusion section has been improved in order to better fit the section on the result discussion.

Lines 426-428

“the as built specimens show a weak sensitivity to strain rate for both combinations of process parameters employed in the experimental campaign, detecting a smaller increment in both DIFfy and DIFfu compared to the as cast ones for strain rates ranging between 200 and 800 1/s;”

Lines

“the increase in UTS and decrease in elongation at break detected for the as built specimens at 400 W can be explained by a higher presence of inclusions compared to the as built specimens at 300 W. Nevertheless, further analyses are required to properly characterize the mechanical behavior of the as built samples and a microstructural characterization is currently running in order to identify and quantify the metallic inclusions detected by optical imaging;”

Round 3

Reviewer 1 Report

The authors have revised the manuscript accoding to the suggestions. A small suggestion is that 

The unit “ 1/s” should be “ /s”

Author Response

Dear reviewer,

the Authors would like to thank you for your valuable comments and suggestions. We hope to have properly responded to your comments and to have significantly improved the quality of our Manuscript.

Please, find the answers to your kind comments as follows.

Thank you

Point 1: The authors have revised the manuscript according to the suggestions. A small suggestion is that:

The unit “1/s” should be “/s”

Response 1: Thank you for your comment and suggestions. The unit “1/s” has been replaced with “/s” all through the Manuscript.
